# The association between frailty, care receipt and unmet need for care with the risk of hospital admissions

Asri Maharani[1]*, David R. Sinclair[2], Andrew Clegg[3], Barbara Hanratty[2], James Nazroo[4], Gindo Tampubolon[5], Chris Todd[1], Raphael Wittenberg[6], Terence W. O'Neill[1‡], Fiona E. Matthews[2‡]

1 National Institute for Health and Care Research (NIHR) Policy Research Unit in Older People and Frailty / Healthy Ageing, School of Health Sciences, Faculty of Biology, Medicine and Health, The University of Manchester, Manchester, United Kingdom, 2 National Institute for Health and Care Research (NIHR) Policy Research Unit in Older People and Frailty / Healthy Ageing, Population Health Sciences Institute, Newcastle University, Newcastle-upon-Tyne, United Kingdom, 3 Academic Unit for Ageing and Stroke Research, Bradford Institute for Health Research, School of Medicine, University of Leeds, Leeds, United Kingdom, 4 Cathie Marsh Institute for Social Research, School of Social Sciences, Faculty of Humanities, University of Manchester, Manchester, United Kingdom, 5 Global Development Institute, University of Manchester, Manchester, United Kingdom, 6 National Institute for Health and Care Research (NIHR) Policy Research Unit in Older People and Frailty / Healthy Ageing, Care Policy and Evaluation Centre, London School of Economics and Political Science, London, United Kingdom

☯ These authors contributed equally to this work.
‡ TWO and FEM also contributed equally to this work.
* asri.maharani@manchester.ac.uk

## Abstract

### Background

Frailty is characterised by a decline in physical, cognitive, energy, and health reserves and is linked to greater functional dependency and higher social care utilisation. However, the relationship between receiving care, or receiving insufficient care among older people with different frailty status and the risk of unplanned admission to hospital for any cause, or the risk of falls and fractures remains unclear.

### Methods and findings

This study used information from 7,656 adults aged 60 and older participating in the English Longitudinal Study of Ageing (ELSA) waves 6–8. Care status was assessed through received care and self-reported unmet care needs, while frailty was measured using a frailty index. Competing-risk regression analysis was used (with death as a potential competing risk), adjusted for demographic and socioeconomic confounders. Around a quarter of the participants received care, of which approximately 60% received low levels of care, while the rest had high levels of care. Older people who received low and high levels of care had a higher risk of unplanned admission independent of frailty status. Unmet need for care was not significantly associated with an increased risk of unplanned admission compared to those receiving no care. Older people in receipt of care had an increased risk of hospitalisation due to falls but not fractures, compared to those who received no care after adjustment for covariates, including frailty status.

**Data Availability Statement:** Data are available in a public, open access repository. ELSA data from the main survey (SN 5050), and the COVID-19 substudy (SN 8688), are available through the UK

Data Service (https://ukdataservice.ac.uk/). Details on how to access ELSA, including the conditions of use, can be found on the ELSA website (https://www.elsa-project.ac.uk/accessing-elsa-data) and the UK Data Service website.

**Funding:** This research was funded through the National Institute for Health and Care Research (NIHR) Policy Research Unit in Older People and Frailty (funding reference PR-PRU-1217-2150). As of 01.01.24, the unit has been renamed to the NIHR Policy Research Unit in Healthy Ageing (funding reference NIHR206119). The views expressed are those of the author(s) and not necessarily those of the NIHR or the Department of Health and Social Care.

**Competing interests:** The authors have declared that no competing interests exist.

## Conclusions

Care receipt increases the risk of hospitalisation substantially, suggesting this is a group worthy of prevention intervention focus.

## Introduction

Demand for care services for older people is increasing as the global population continues to age [1]. The World Health Organisation (WHO) Global Strategy and Action Plan on Ageing and Health 2016–2020 highlighted the right of older people to receive care and support to maintain their best possible functional abilities [2]. Frailty, which describes how we gradually lose our in-built reserves with increasing age [3], is a framework for understanding health discrepancies among older adults and a significant predictor of care receipt [4]. Estimates of frailty prevalence worldwide vary between 12% to 24% [5]. Almost all older people with frailty (93%) experience mobility problems, and over half of them have difficulties with washing, dressing or housework [6]. Older people with frailty are thus more likely to be in need of social care services.

A prior study estimated that caring for frail people will cost between 4 and 9 times as much as caring for healthy people [7]. In the UK, social care is provided through paid care from public and private funding and unpaid care from friends and family. Despite this, a report estimated that 1.5 million people over 65 in England have unmet care needs [8]. Our prior work estimated that around 0.7 million and 1.6 million people aged 65+ in England were frail and prefrail in 2018, respectively. However, only 0.5 million adults in the same age group received government funding for care [9]. We also found that 82% (124 from 151) of the local authorities in the study had a greater number of persons with frailty aged 65+ than care recipients within the same age range, suggesting, given that frail individuals are more likely to require care, that there is a care deficit present in much of the country.

Frailty is associated with increased healthcare use, and hospital admissions represent a substantial proportion of the overall costs associated with the condition [10, 11]. Frailty is associated with an annual additional 1.0 million emergency admissions and 1.1 million elective admissions in England [12]. Frail patients are also more likely to be attended by an ambulance for incidents which do not require conveyance to a hospital [13]. In addition, severely frail older people have seven times longer lengths of stay in hospital following emergency hospitalisation than non-frail older people. The negative consequences of unmet care needs among older people on mental health problems [14, 15] and higher mortality rates [16] have been documented in the literature. However, there is limited evidence on the effect of care receipt and unmet need for care among older people with different frailty status and their future healthcare utilisation.

This study aimed to understand how care receipt and unmet need for care among older people with different frailty status are associated with the risk of unplanned admission to the hospital for any cause and for conditions associated with frailty, specifically falls [17, 18] and fractures [19, 20].

## Materials and methods

### Participants and setting

The analysis uses a dataset that combines the English Longitudinal Study of Ageing (ELSA) [21] with the census of public hospital records in England, the Hospital Episode Statistics

(HES) [22], and mortality data from the Office for National Statistics (ONS) [23]. ELSA is a panel survey of a representative sample of the household population aged 50+ in England [21]. ELSA waves are performed every two years, collecting information on demographic, socioeconomic, and health characteristics. To date, it has conducted nine waves. Our analysis used data from ELSA waves 6 to 8, covering 2012–2017, as the information on the types of care received is available from wave 6, and HES and ONS data were available until 31 January 2018. All individuals included in the analysis had data linked to HES and ONS mortality (including those who dropped out of the study after the baseline survey). In this study, we included ELSA participants aged 60 and older.

## Measures

**Frailty.** Frailty was assessed using a frailty index derived from data collected as part of ELSA. The frailty index included 60 variables ('deficits') representing conditions that accumulate with age and are associated with adverse outcomes, including disability, mobility, sensory impairments, cognitive function, and chronic diseases. The full list of variables used to create the frailty index is shown in **S1 Table**. An individual's frailty index is calculated as the proportion of possible deficits present in an individual. Frailty indices with at least 30–40 deficits can predict adverse outcomes accurately [24, 25]. Frailty was measured at baseline (Wave 6). We categorised the frailty index into three groups: robust (frailty index ≤ 0.08), prefrail (frailty index >0.08–0.25) and frail (frailty index ≥ 0.25) [26].

**Level of care and unmet need for care.** Respondents in ELSA were asked to respond to questions about their care if they reported having at least one difficulty with mobility, an Activity of Daily Living (ADL) or an Instrumental Activity of Daily Living (IADL) [27]. Based on the level of care received, we categorised respondents into those in receipt of: (1) high levels of care, if the respondents received help in the last month for using the toilet, getting in and out, eating, bathing/showering, walking across a room, dressing, and having meals on wheels; (2) low levels of care, if the respondents received help in the last month for grocery shopping, house or garden work, managing money, climbing at least one flight of stairs without resting, taking medication, walking 100 yards and if they had attended a day centre; and (3) did not receive care.

Participants who have received care were also asked whether their care meets their needs. We classified the respondents into having: (1) unmet care needs, if they answered that the care they sometimes had or hardly met their needs; and (2) met care needs, if they answered that the care they had met or usually met their needs; and (3) did not receive care.

**Outcome measures.** Unplanned admissions were derived from the HES data linked by NHS Digital to ELSA participants' NHS number, date of birth, gender and postcode. An unplanned admission was defined as admission to the hospital through (1) accident and emergency (A&E); (2) general practitioner (GP) after request of immediate admission; (3) bed bureau [28]; (4) consultant clinic; (5) Mental Health Crisis Resolution team; and (6) other A&E [29]. The full list of the HES method of admission codes is shown in **S2 Table** [29].

Hospitalisation due to falls was defined as the first hospitalisation where a diagnosis of fall was recorded since baseline (wave 6) based on the International Classification of Disease 10th version (ICD-10) of falls, i.e., W00 to W19 [30, 31]. Hospitalisation due to fractures was the first hospitalisation where a fracture diagnosis was recorded since the baseline corresponded to the ICD-10 M, S and T codes (see **S3 Table**).

**Covariates.** Age was included in the principal analysis as a continuous variable and in sensitivity analysis after categorisation into 5-year age groups (60–64; 65–69; 70–74; 75–79; 80–84; 85+). Gender (male/female), ethnicity (white/non-white) and marital status (married/

not married) were categorised as indicated. Educational attainment was categorised into lower than secondary school (reference), secondary school, and college or higher. Wealth was measured by the net total wealth of the respondent's benefit unit (defined as a single adult, or a married or cohabiting couple, and any dependent children [32]). Net total wealth comprised the sum of savings and investments after subtracting financial debt. We split wealth into quintiles to investigate the hierarchical effects of wealth.

## Statistical analysis

To examine the effect of the mismatch between levels of frailty and receipt of care on each hospitalisation category in this study, we employed competing-risk regression analysis using a version of the Fine and Gray analysis [33]. This analysis allows a competing risk–an event that might occur during the follow-up instead of the event of interest–to be considered in the model. Death is the potential competing risk in this study when examining hospital admissions. Mortality status was ascertained from linked register data up to the end of January 2018. Frailty, level of care and need for care were defined in wave 6 (2012/2013) and the follow-up time up to 31 January 2018. We present the results as the subdistribution hazard ratios (SHRs) and 95% confidence intervals (95% CIs) [34]. The subdistribution hazard function is defined as the instantaneous rate of occurrence of hospitalisation in older people who have not yet experienced it during the study [34]. The SHR is the ratio of these functions in the presence of two different values of a covariate (e.g., a person who is frail relative to a person who is not frail).

For unplanned admissions as the outcome, we performed the analysis separately for the level of care and need for care. The first analysis included frailty status (robust as the reference, prefrail, and frail) and level of care (no care as the reference, low and high levels of care), while the second analysis included frailty status (robust as the reference, prefrail, and frail) and need for care (no care as the reference, met care needs, and unmet care needs). All analyses were adjusted for age, gender, ethnicity, marital status, wealth and education.

We further performed the analysis by gender and categorised the care receipt into: (1) received care; and (2) did not receive care. The same categorisation was used to analyse conditions associated with frailty: falls and fractures.

We checked for the presence of an interaction between frailty status, level of care, and need for care by creating a second model for each analysis. In Model 2, we created nine main dependent variables combining frailty status and level of care: (1) robust and received no care (reference group); (2) robust and received low levels of care; (3) robust and received high levels of care; (4) prefrail and received no care; (5) prefrail and received low levels of care; (6) prefrail and received high levels of care; (7) frail and received no care; (8) frail and received low levels of care; and (9) frail and received high levels of care. For analysis of the need for care, we created eight main dependent variables combining frailty status and need for care (there were no robust respondents reporting the unmet need for care needs): (1) robust and received no care (reference group); (2) robust and received care; (3) prefrail and received no care; (4) prefrail and reported having met care needs; (5) prefrail and reported having unmet care needs; (6) frail and received no care; (7) frail and reported having met care needs; and (8) frail and reported having unmet care needs. We looked for an interaction between frailty status, level of care and need for care on the risk of hospitalisation by plotting the SHRs and 95% CIs using both models. In order to compare Model 1 (without interaction) with Model 2 (with interaction), we calculated the SHRs of each category (i.e., robust and received no care as the reference; robust and received low levels of care; robust and received high levels of care; prefrail and received no care; prefrail and received low levels of care; prefrail and received high levels

of care; frail and received no care; frail and received low levels of care; and frail and received high levels of care) by adding the log of each frailty status, level of care, and need for care and then taking its exponential. An interaction effect was considered to exist if the two plots showed different values of the association of the categories and the risk of hospitalisation. **S1** and **S2** **Figs** show that the two plots have similar values, suggesting no interaction between frailty status and care receipt in their relationships with the risk of hospitalisation. The model without an interaction was thus preferable. Survey data was weighted using ELSA cross-sectional survey weight at wave 6.

**Sensitivity analysis.**　We performed three types of sensitivity analyses. Firstly, we used age categorised into groups (60–64; 65–69; 70–74; 75–79; 80–84; 85+) instead of age as a continuous variable.

Secondly, we performed two analyses on different sets of short epochs of time. The first set of epochs of time are: (1) wave 6 as the baseline with 6 months follow-up; (2) wave 7 baseline with 6 months follow-up; and (3) wave 8 baseline; 6 months follow-up. The second set of epochs of time are: (1) wave 6 baseline with 12 months follow-up; (2) wave 7 baseline with 12 months follow-up; and (3) wave 8 baseline; 6 months follow-up. We performed two meta-analyses using those two sets of epochs of time. In those analyses, frailty status, level of care and need for care were defined at each wave 6, 7, and 8. The start date was defined as the interview date. Age was defined as the age at each wave, and we had two different follow-up lengths for each wave, except for wave 8: 6 and 12 months. We could not have a similar follow-up length in wave 8 as the data were only available until 31 January 2018 (6 months after Wave 8 enrolled).

Finally, we performed the analysis by putting a censor date between two interview dates if there were any changes in frailty status, level of care or need for care between the two waves of ELSA. When a person's response changed between waves, we assumed the change occurred midway between the waves (censor date). The respondents were followed up until the censor date, death or end of the study if they did not change frailty status.

## Results

### Participant characteristics

Descriptive characteristics of the study sample are presented in **Table 1**. A total of 7,656 participants, 3,535 men and 4,121 women, were included in the analysis. The mean age was 71.1 years. The majority (97.2%) were white and 65.3% were married. Almost half (48.8%) of the respondents graduated from college or higher education level. After applying sample weighting, the proportion of participants who were frail and prefrail was estimated as 17.7% and 40.6%, respectively.

The proportion of respondents with pre-frailty and frailty increased with age. Almost 10% of people aged 60–64 were frail, increasing to 44.4% among those aged 85+. Compared to men, women were more likely to be frail (20.5% vs 14.5%) and prefrail (43.9% vs 36.7%). Compared to those who did not complete high school, people who graduated from high school and college or higher were less likely to be frail and prefrail. The proportion of respondents with frailty increased from 5.4% among the wealthiest quintile to 28.7% among the least wealthy quintile.

Around a quarter of adults aged 60+ in England received care, of which approximately 60% received low levels, while the rest had high levels of care. The level of care receipt is proportionally higher among frail and prefrail than robust older people: 6.4% and 47.6% of the prefrail and frail respondents received high levels of care, respectively. Around a fifth (20.9%) of respondents with prefrailty received low levels of care, while 36.0% of those with frailty had

**Table 1. Descriptive characteristics of the respondents at baseline.**

| | Total* | Robust** | Prefrail** | Frail** |
|---|---|---|---|---|
| Frailty index, mean (SD) | 0.1 (0.1) | | | |
| *Frailty status, n (%)* | | | | |
| Robust | 3,357 (43.9) | 2,910 (41.7) | | |
| Prefrail | 3,026 (39.5) | | 2,833 (40.6) | |
| Frail | 1,268 (16.6) | | | 1,239 (17.7) |
| Age, mean (SD) | 71.1 (8.2) | 68.10 (6.5) | 72.73 (8.4) | 76.28 (9.6) |
| *Sex, n (%)* | | | | |
| Males | 3,535 (46.2) | 1,574 (48.8) | 1,182 (36.7) | 468 (14.5) |
| Females | 4,121 (53.8) | 1,336 (35.6) | 1,651 (43.9) | 771 (20.5) |
| *Ethnicity, n (%)* | | | | |
| White | 7,442 (97.2) | 2,819 (41.8) | 2,758 (40.9) | 1,169 (17.3) |
| Non-White | 214 (2.8) | 91 (38.6) | 75 (31.9) | 69 (29.5) |
| *Married, n (%)* | | | | |
| No | 2,653 (34.7) | 696 (28.6) | 1,065 (43.7) | 677 (27.8) |
| Yes | 5,001 (65.3) | 2,213 (48.7) | 1,767 (38.9) | 562 (12.4) |
| *Education attainment, n (%)* | | | | |
| Less than secondary school | 2,507 (32.7) | 706 (28.1) | 1,108 (44.1) | 699 (27.8) |
| Secondary school | 1,414 (18.5) | 570 (45.4) | 528 (42.0) | 159 (12.6) |
| College or higher | 3,735 (48.8) | 1,634 (50.9) | 1,197 (37.3) | 381 (11.9) |
| *Wealth, n (%)* | | | | |
| 5th quintile (most wealthy) | 1,500 (20.0) | 859 (61.6) | 460 (33.0) | 75 (5.4) |
| 4th | 1,500 (20.0) | 686 (49.1) | 608 (43.5) | 103 (7.4) |
| 3rd | 1,499 (20.0) | 614 (44.1) | 604 (43.3) | 177 (12.7) |
| 2nd | 1,504 (20.0) | 498 (35.7) | 617 (44.2) | 281 (20.1) |
| 1st quintile (least wealthy) | 1,495 (19.9) | 334 (23.9) | 661 (47.3) | 401 (28.7) |
| *Level of care received, n (%)* | | | | |
| No care | 5,213 (74.0) | 2,869 (55.0) | 2,154 (41.3) | 190 (3.7) |
| Receiving low levels of care | 1,080 (15.3) | 43 (4.0) | 620 (57.4) | 417 (38.6) |
| Receiving high levels of care | 749 (10.6) | 6 (0.8) | 190 (25.4) | 553 (73.8) |
| *Need for care, n (%)* | | | | |
| No care | 5,213 (74.0) | 2,869 (55.0) | 2154 (41.3) | 190 (3.7) |
| Met care needs | 1,167 (16.6) | 27 (2.3) | 559 (47.9) | 581 (49.8) |
| Unmet care needs | 539 (7.7) | 7 (1.2) | 170 (31.6) | 362 (67.2) |

Notes

\* unweighted

\*\* weighted.

low levels of care. Characteristics of respondents at baseline by level of care are shown in **S4 Table**. It shows that the proportions of individuals receiving either low or high levels of care (compared to no care) were higher among those who were older, female, non-White, not married, those who had lower educational attainment and who were less wealthy.

Around 16.6% of respondents with prefrailty stated that their care needs were met, while 7.7% reported unmet needs for care. Half of the respondents with frailty stated that they had met care needs, while almost one-third (31.3%) reported unmet care needs. Characteristics of respondents at baseline categorised by the need for care are shown in **S5 Table**. The proportions of individuals reporting unmet need for care were higher among those who were older, female, non-White, not married, had lower education attainment and were less wealthy.

## Frailty status, level of care and risk of unplanned hospital admission

During five years of follow-up, there were 2,663 unplanned admissions and 310 deaths (**S6 Table**). In an unadjusted competing risk model, compared to those who were robust, the sub-distribution hazard ratios (SHRs) for unplanned hospital admission among people who were prefrail and frail were 1.80 (95%CI: 1.64; 1.97) and 2.74 (95%CI: 2.47; 3.03) respectively, see **S7 Table**. Compared to those who received no care, those who received either low or high levels of care were more likely to have an unplanned hospital admission: SHR 1.70 (95%CI:1.55; 1.87) and 1.82 (95%CI:1.64; 2.02) respectively.

After adjustment for covariates, the SHRs for unplanned hospital admission among those who were prefrail and frail were attenuated (see **Table 2**). Compared to those who were robust, the adjusted SHR for unplanned admission for prefrailty was 1.76 (95%CI: 1.59; 1.95) and for frailty 2.46 (95%CI:2.13; 2.84). After adjustment for covariates including frailty status, compared to those not receiving care, the adjusted SHR for unplanned admission for those with low levels of care was 1.19 (95%CI:1.06; 1.33) and for those with high levels of care was 1.29 (95%CI:1.12; 1.48).

Taking account of death as a competing risk, the cumulative incidence of unplanned hospital admissions increased over time for all frailty categories; the slope was greater among those who were frail and prefrail than those who were robust (see **Fig 1A**). The slope was also greater within frailty categories for those who received care than those who did not. The cumulative incidence curve for frail people with high levels of care increased steeply with time, followed by frail people with low levels of care.

## Frailty status, need for care and risk of unplanned hospital admission

In an unadjusted competing risk model, compared to those who were not in receipt of care, the SHRs for unplanned hospital admission among people who were in receipt of care and whose care needs were met was 1.82 (95%CI:1.64; 2.02), whilst for those with an unmet need of care the SHR was 2.07 (95%CI:1.61; 2.67), see **S7 Table**.

After adjustment for covariates, including frailty status, the strength of the SHRs was attenuated. Compared to those not receiving care, the adjusted SHR for unplanned admission for those in receipt of care and whose care needs were met was 1.22 (95%CI: 1.09; 1.35), with a similar SHR for unmet need for care 1.21 (95%CI: 0.91; 1.61), though with the confidence interval embracing unity, see **Table 2**.

**Table 2. Subdistribution hazard ratio (95% CI) for the association between frailty status, level of care received, need for care and unplanned admissions.**

| | Level of care | Need for care |
|---|---|---|
| *Frailty status, reference: robust* | | |
| Prefrail | 1.76 (1.59; 1.95) | 1.77 (1.60; 1.95) |
| Frail | 2.46 (2.13; 2.84) | 2.51 (2.18; 2.89) |
| *Level of care received, reference: no care* | | |
| Receiving low levels of care | 1.19 (1.06; 1.33) | |
| Receiving high levels of care | 1.29 (1.12; 1.48) | |
| *Need for care, reference: no care* | | |
| Met care needs | | 1.22 (1.09; 1.35) |
| Unmet care needs | | 1.21 (0.91; 1.61) |

*Note*: Unplanned admissions N = 2,662, competing event deaths N = 310. All models were adjusted for age, gender, marital status, wealth in quintiles and education attainment.

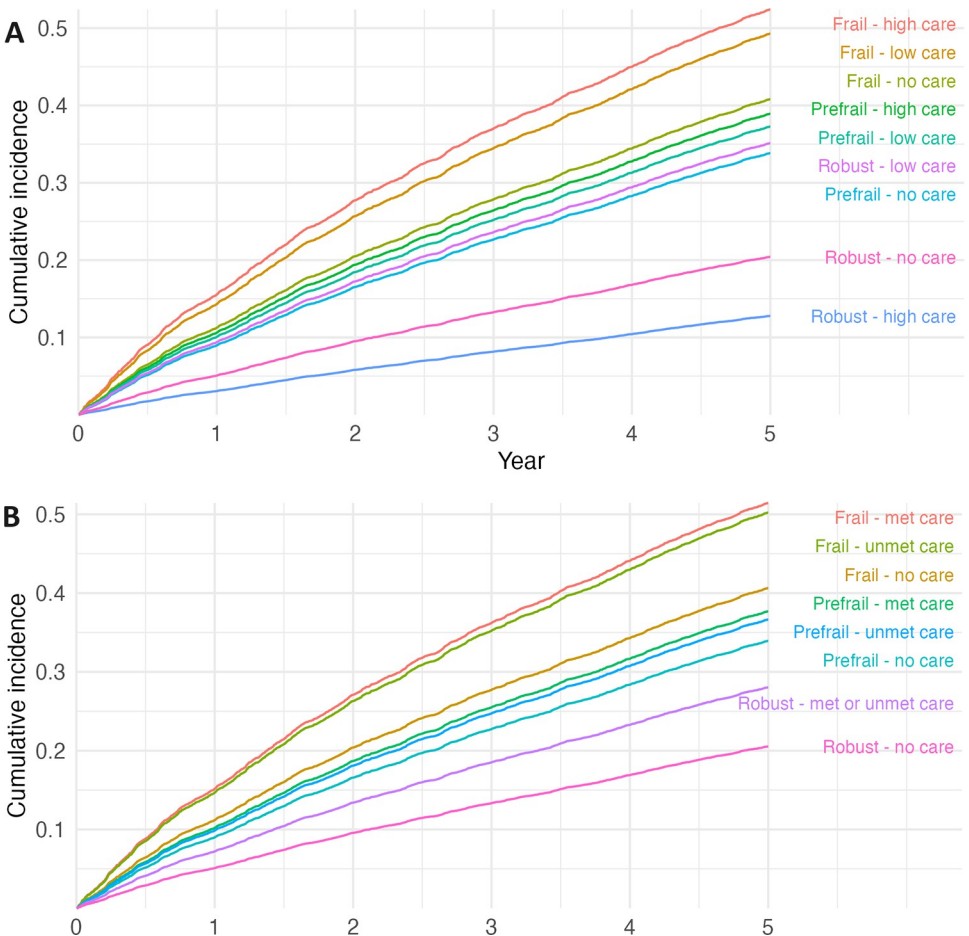

**Fig 1.** Estimates of the cumulative incidence of unplanned hospitalisation according to frailty status and (A) level of care received and (B) need for care. Death was the competing risk.

Taking account of death as a competing risk, the cumulative incidence of unplanned hospital admissions was higher within frailty categories for those who were in receipt of care and whose care needs were met than those with an unmet need for care (**Fig 1B**).

For the first sensitivity analysis, we analysed the interaction between frailty with the level of care and need for care. **S1** and **S2 Figs** show that the analysis of the interaction between frailty with the level of care and need for care have similar values with those excluding the interaction, suggesting no interaction between frailty status and care receipt in their relationships with the risk of hospitalisation. The results of the sensitivity analyses using age group as the covariates (**S8 Table**), five different epochs of time (**S3 Fig**), and varying times of analysis (**S9 Table**) are similar to our principal results, suggesting the results are robust.

## Frailty, level of care and risk of unplanned admission: Influence of gender

Among men, after adjustment for covariates including frailty status, compared to those who received no care, those who received care were associated with an increased risk of unplanned hospitalisation (SHRs 1.30; 95% CI 1.09, 1.54), see **S10 Table**. This was also true for women (SHRs 1.31; 95% CI 1.14, 1.50). **S4 Fig** shows that among men, those who were frail and received care had the steepest estimated cumulative incidence, followed by those who were

**Table 3. Subdistribution hazard ratio (95% CI) for the association between frailty status and care receipt with hospitalisation due to falls and fractures, England 2012–2018.**

| | Hospitalisation due to falls[a] | Hospitalisation due to fractures[a] |
|---|---|---|
| *Frailty status, reference: robust* | | |
| Prefrail | 2.18 (1.68; 2.83) | 1.78 (1.35; 2.34) |
| Frail | 2.73 (1.95; 3.80) | 2.11 (1.45; 3.07) |
| *Received care, reference: No* | | |
| Yes | 1.30 (1.03; 1.63) | 1.25 (0.95; 1.63) |

*Note*: [a]Adjusted for age, gender, ethnicity, marital status, wealth and education.

frail and did not receive care. This order was similar for women, as being frail and receiving care had a steeper estimated cumulative incidence of frail.

## Frailty status, receipt of care and the risk of admissions due to falls and fractures

During five years of follow-up, there were 586 admissions due to falls and 432 admissions due to fractures (**S6 Table**). **Table 3** reports the SHR for the association between frailty and care receipt levels and the risk of hospitalisation due to a fall estimated using competing risk analysis. The adjusted SHRs for hospitalisation due to a fall among older adults who were prefrail and frail were 2.18 (95%CI: 1.68; 2.83) and 2.73 (95%CI: 1.95; 3.80), respectively, compared with those who were robust. Receiving care was associated with a 1.30 (95% CI: 1.03; 1.63) higher risk of admissions due to falls.

The adjusted SHRs for hospitalisation due to a fracture among older adults who were prefrail and frail were 1.78 (95%CI: 1.35; 2.34) and 2.11 (95%CI: 1.45; 3.07), respectively, compared with those who were robust. Receiving care (SHR: 1.25; 95% CI: 0.95; 1.63) was not significantly associated with an increased risk of admissions due to fractures.

**Fig 2A** shows that frail older people had the steepest estimated cumulative incidence curves for hospitalisation due to falls, followed by those who were prefrail and robust. For fractures, prefrail older people with care had the steepest estimated cumulative incidence curve, followed by frail older people with no care (**Fig 2B**). In both cases (falls -2A and fractures -2B), the estimated cumulative incidence curves for hospitalisation were the steepest for frail older people regardless of whether or not they received care. It is also noticeable that each reason for admission (falls and fractures) for each group (frail, prefrail, robust) receiving care always fares worse than those not receiving care.

## Discussion

Using a large population-based survey (ELSA) linked to national hospitalisation and mortality records, we found that 15.2% and 10.4% of adults aged 60+ in England received low and high levels of care, respectively, with the proportion reporting care receipt higher among prefrail and frail than robust individuals. The data are consistent with previous findings [35–37]. For instance, a study based on primary care in Norwich found that the average number of care plans required per referral was higher among severely frail older patients (2.97) than fit patients (2.22), indicating more complex care needs in the community [36]. In a cross-sectional study in the Netherlands, frail older adults with more ADL limitations and a higher frailty score were more likely to have higher care needs [38].

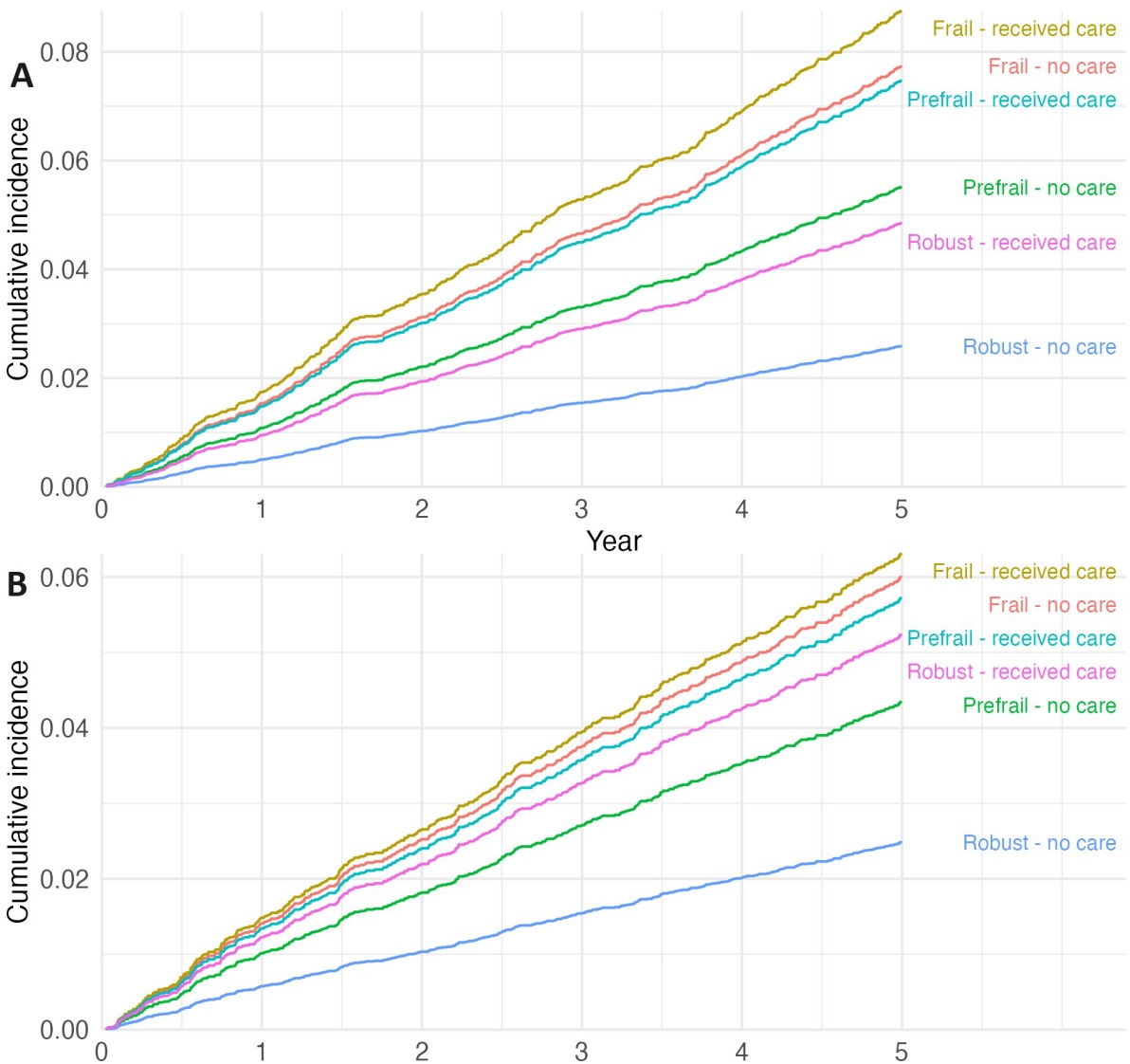

**Fig 2. Estimates of the cumulative incidence curves of risk of hospitalisation due to falls and fractures according to frailty status and receipt of care.** Death was the competing risk. (A) Hospitalisation due to falls; (B) Hospitalisation due to fractures.

Our results suggest that compared to those receiving no care, receiving low or high levels of care was associated with a higher risk of unplanned admission and hospital admissions due to falls independent of frailty status. The finding may suggest the presence of other factors relating to falls were not captured by the frailty index, including Parkinson's disease [39], history of falls [40, 41], and polypharmacy [42]. Future studies may include these factors in predicting medical care usage. Another factor which may affect hospitalisation is living status. The risk of falls might be higher among older people living alone [43] because of the amount of time between carer visits, no one around to help with the toilet, and concern that it is not a 'safe' environment to leave someone in post-fall.

In our analysis, the proportion of unmet care needs was highest among frail older people. An unmet need for care was associated with a small though non-significant risk of unplanned hospitalisation, with the magnitude of the risk similar to those whose care needs were reported

as being met. However, caution is needed in interpreting these data as our definition of care needs focuses on the *adequacy* (met / unmet) of those who were already receiving care. There is a relative lack of data concerning the role of the unmet need for care as a contextual factor when examining frailty and adverse health outcomes in older adults, for which further research is needed. Supporting our finding, data from a Canadian study suggest that perceived unmet need for care among adults with chronic conditions was not associated with an increased risk of hospital admission [44], while two American studies did find an association [45, 46].

We found that 40.6% and 17.7% of adults aged 60+ in England were prefrail and frail, respectively. Both frailty and prefrailty (compared to being robust) were associated with a higher risk of unplanned hospital admission and hospital admissions due to falls and fractures after adjusting with care receipt and unmet need for care. These findings corroborate previous studies that report an association between frailty and an increase in emergency and elective hospital admissions [12, 35, 36, 47]. The impact of frailty on healthcare utilisation is substantial: the length of inpatient stay for severely frail patients was seven times longer than for non-frail patients [12]. In relation to the influence of gender, our data suggest that after adjusting for covariates, receiving care (compared to receiving no care) was associated with a higher risk of unplanned admissions among men and women and with a magnitude of risk similar in men and women.

Strengths of our analysis include the nationally representative sample of non-institutional-ised individuals, which is generalisable to the English population. Furthermore, the survey used in this study was linked to national hospitalisation and mortality data, which minimised loss at follow-up. Additionally, this study used a competing risk analysis strategy to consider mortality as a competing event rather than a survival analysis. Competing risk analysis accommodates the competing nature of multiple causes of the same event.

Several limitations need to be considered in interpreting the findings. First, care receipt and need for care were measured only at baseline, with no follow-up data. It was not possible, therefore, to address how changes in care receipt and care needs may have affected hospitalisa-tion among older people. Second, questions about care were only asked when a respondent reported having difficulties in mobility, ADL or IADL in ELSA. Thus, information on care receipt and the need for care excluded those who did not report any functional difficulties; it is possible that more people would have reported care receipt and care needs if the entire sample had been asked. In addition, perceived unmet needs were measured using only one question in ELSA, which did not distinguish between different care needs. A cross-sectional study among frail older adults in the Netherlands examined different types of unmet care needs, i.e., environmental (accommodation, household activities, food, and caring for another), physical needs (physical health, medication use, visual/hearing impairment, mobility/falls, and self-care), and psychosocial needs (memory, company, daytime activities, and information) [38]. The respondents reported the highest proportion of unmet care needs in the psychosocial domain. It is possible that different types of unmet needs may affect adverse health outcomes differently. Finally, the frailty index constructed in this study did not include the diagnosis of sarcopenia and nutritional status due to the unavailability of the information in ELSA. The Italian frailty index, for instance, includes the nutritional index and provides good reliability and validity in predicting mortality, disability and hospitalisation [48]. Future research may include sarcopenia, nutritional status, and other geriatric assessments in constructing a frailty index to allow for a more comprehensive assessment of an older adult's health.

Our findings have potential implications. In our analysis, frailty was associated with an increased risk of unplanned admission to the hospital. As frailty is a potentially reversible health state [49], early screening and intervention, good-quality and timely diagnosis of pre-frailty and frailty in the community, and effective interventions at an early stage could be

effective strategies for reducing or delaying the utilisation of secondary care services. Prior study shows that low social support is associated with long-term mortality among older people [50]. Our data suggest that older people with frailty or prefrailty who are *already in receipt of care* are at significantly greater risk of unplanned hospitalisation and, therefore, a group who may potentially benefit from more detailed assessment and targeted or personalised community-based interventions with the aim of reducing their risk.

In conclusion, older men and women who are in receipt of care are at increased risk of unplanned hospitalisation and other adverse outcomes. Those who are frail or prefrail are at greater risk of hospitalisation, providing opportunities for targeted community-based interventions to reduce the impact on already overstretched secondary care services.

## Supporting information

**S1 Fig. Comparison between Model 1 and Model 2 in identifying the association between frailty status and level of care with unplanned admissions.**
(TIF)

**S2 Fig. Comparison between Model 1 and Model 2 identifying the association between frailty status and need for care with unplanned admissions.**
(TIF)

**S3 Fig. Subdistribution hazard ratio (95% CI) for the association between frailty status and level of care with unplanned admissions in as the determinant in each epoch of time (particular period of time).**
(TIF)

**S4 Fig. Estimates of the cumulative incidence curves of risk of unplanned hospitalisation according to frailty status and receipt of care by gender.** Death was the competing risk.
(TIF)

**S1 Table. Deficit variables included in the ELSA frailty index.**
(DOCX)

**S2 Table. Hospital Episode Statistics–Method of admission categories.**
(DOCX)

**S3 Table. ICD-10 codes for fractures.**
(DOCX)

**S4 Table. Descriptive characteristics of the respondents (n = 6,984) by level of care in ELSA wave 6 (2012/2013).**
(DOCX)

**S5 Table. Descriptive characteristics of the respondents (N = 6,984) by need for care in ELSA wave 6 (2012/2013).**
(DOCX)

**S6 Table. The number of hospital admissions and death in each outcome.** Presented are number (%).
(DOCX)

**S7 Table. Unadjusted subdistribution hazard ratio (95% CI) for the association between frailty status, level of care, need for care and each of the covariates with unplanned admissions.** Unplanned admissions N = 2,662, competing event deaths N = 310.
(DOCX)

**S8 Table. Subdistribution hazard ratio (95% CI) for the association between frailty status, frequency for care and need for care with unplanned admissions with age group as the determinant.**
(DOCX)

**S9 Table. Subdistribution hazard ratio (95% CI) for the association between frailty status, level of care, and need for care with unplanned admissions with varying time analysis.** Adjusted for age group, gender, ethnicity, marital status, wealth and education.
(DOCX)

**S10 Table. Subdistribution hazard ratio (95% CI) for the association between frailty status and receiving care with unplanned admissions by gender.** Unplanned admissions N = 2,662, competing event deaths N = 310. Adjusted for age group, gender, ethnicity, marital status, wealth and education.
(DOCX)

## Acknowledgments

We thank our academic and professional support colleagues from the National Institute for Health and Care Research Policy Research Unit in Older People and Frailty / Healthy Ageing, with whom we discussed the ideas presented in this paper during unit meetings.

## Author Contributions

**Conceptualization:** Asri Maharani, David R. Sinclair, Terence W. O'Neill, Fiona E. Matthews.

**Data curation:** Asri Maharani, David R. Sinclair.

**Formal analysis:** Asri Maharani, David R. Sinclair.

**Funding acquisition:** Barbara Hanratty, Chris Todd.

**Methodology:** Asri Maharani, David R. Sinclair, Terence W. O'Neill, Fiona E. Matthews.

**Supervision:** Andrew Clegg, Barbara Hanratty, James Nazroo, Chris Todd, Terence W. O'Neill, Fiona E. Matthews.

**Validation:** Andrew Clegg, Barbara Hanratty, James Nazroo, Chris Todd, Terence W. O'Neill, Fiona E. Matthews.

**Visualization:** Asri Maharani, David R. Sinclair.

**Writing – original draft:** Asri Maharani, David R. Sinclair, Terence W. O'Neill, Fiona E. Matthews.

**Writing – review & editing:** Asri Maharani, David R. Sinclair, Andrew Clegg, Barbara Hanratty, James Nazroo, Gindo Tampubolon, Chris Todd, Raphael Wittenberg, Terence W. O'Neill, Fiona E. Matthews.

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
