## [Decision Letter · Decision Letter 0]

23 May 2024

PONE-D-24-05107The association between frailty, care receipt and unmet need for care with the risk of hospital admissionsPLOS ONE

Dear Dr. Maharani,

Thank you for submitting your manuscript to PLOS ONE. After careful consideration, we feel that it has merit but does not fully meet PLOS ONE’s publication criteria as it currently stands. Therefore, we invite you to submit a revised version of the manuscript that addresses the points raised during the review process.

We look forward to receiving your revised manuscript.

Kind regards,

Pasquale Abete

Academic Editor

PLOS ONE

Journal Requirements:

“This research was funded through the National Institute for Health and Care Research (NIHR) Policy Research Unit in Older People and Frailty (funding reference PR-PRU-1217-2150). As of 01.01.24, the unit has been renamed to the NIHR Policy Research Unit in Healthy Ageing (funding reference NIHR206119). The views expressed are those of the author(s) and not necessarily those of the NIHR or the Department of Health and Social Care.”

Additional Editor Comments:

According to Reviewers' decision, the manuscript needs a minor revision.

Reviewers' comments:

Reviewer's Responses to Questions

**Comments to the Author**

1. Is the manuscript technically sound, and do the data support the conclusions?

Reviewer #1: Yes

Reviewer #2: Yes

2. Has the statistical analysis been performed appropriately and rigorously? 

Reviewer #1: Yes

Reviewer #2: Yes

3. Have the authors made all data underlying the findings in their manuscript fully available?

Reviewer #1: Yes

Reviewer #2: Yes

4. Is the manuscript presented in an intelligible fashion and written in standard English?

Reviewer #1: Yes

Reviewer #2: Yes

5. Review Comments to the Author

Reviewer #1: This study used information from 7,656 adults aged 60 and older participating in the English Longitudinal Study of Ageing (ELSA, waves 6-8). Care status was assessed through received care and self-reported unmet care needs, while frailty was measured using a frailty index. Competing-risk regression analysis was used (with death as a potential competing risk), adjusted for demographic and socioeconomic confounders. Around a quarter of the participants received care, of which approximately 60% received low levels of care, while the rest had high levels of care. Older people who received low and high levels of care had a higher risk of unplanned admission independent of frailty status. Unmet need for care was not significantly associated with an increased risk of unplanned admission compared to those receiving no care. Older people in receipt of care had an increased risk of hospitalization due to falls but not fractures, compared to those who received no care after adjustment for covariates, including frailty status. Conclusions: Care receipt increases risk of hospitalization substantially, suggesting this is a group worthy of prevention intervention focus. The manuscript is interesting. However, I have a couple of questions about the frailty measurements. In Frailty index used in the present study, sarcopenia and nutritional status do not seem to be considered. It should be a limitation of the study. In frailty evaluation, both parameters are frequently included in the frailty assessment tool. Please see and discuss Abete P et al. The Italian version of the "frailty index" based on deficits in health: a validation study. Aging Clin Exp Res. 2017 Oct;29(5):913-926.

Reviewer #2: This study aimed to evaluate how care receipt and unmet need for care among older people with different frailty status are associated with the risk of unplanned admission to the hospital for any cause and for conditions associated with frailty, specifically falls and fractures. This study used information from 7,656 adults aged 60 and older participating in the English Longitudinal Study of Ageing (ELSA) waves 6-8. Care status was assessed through received care and self-reported unmet care needs, while frailty was measured using a frailty index. Competing-risk regression analysis was used (with death as a potential competing risk), adjusted for demographic and socioeconomic confounders. Around a quarter of the participants received care, of which approximately 60% received low levels of care, while the rest had high levels of care. Older people who received low and high levels of care had a higher risk of unplanned admission independent of frailty status. Unmet need for care was not significantly associated with an increased risk of unplanned admission compared to those receiving no care. Older people in receipt of care had an increased risk of hospitalization due to falls but not fractures, compared to those who received no care after adjustment for covariates, including frailty status.

The study is based on a large sample size and information derived from the study are relevant for English health system. The continuity of care should be ensured by community care intervention, and I’m absolutely agree that intervention should be based on several factors such as comorbidity, frailty status and social support. [Mazzella, F., Cacciatore, F., Galizia, G., Della-Morte, D., Rossetti, M., Abbruzzese, R., et al. (2010). Social support and long-term mortality in the elderly: role of comorbidity. ARCHIVES OF GERONTOLOGY AND GERIATRICS, 51(3), 323-328] The unmet need (social and medical) is probably one of the main determinants on quality of life and appropriate health services use.

I found the study of interest. Tables should be improved and simplified. I suggest that it might be beneficial to consider adding more information to the figure legend.

6. PLOS authors have the option to publish the peer review history of their article (what does this mean?). If published, this will include your full peer review and any attached files.

Reviewer #1: No

Reviewer #2: **Yes: **cacciatore francesco

---

## [Author Response · Author response to Decision Letter 0]

12 Jun 2024

Comments from Editor

1. Please ensure that your manuscript meets PLOS ONE's style requirements, including those for file naming. The PLOS ONE style templates can be found at https://journals.plos.org/plosone/s/file?id=wjVg/PLOSOne_formatting_sample_main_body.pdf [journals.plos.org] and https://journals.plos.org/plosone/s/file?id=ba62/PLOSOne_formatting_sample_title_authors_affiliations.pdf [journals.plos.org]

Authors’ response

Thank you for the comments. We have ensured that our manuscript meets PLOS ONE’s style requirements.

Comments from Editor

“This research was funded through the National Institute for Health and Care Research (NIHR) Policy Research Unit in Older People and Frailty (funding reference PR-PRU-1217-2150). As of 01.01.24, the unit has been renamed to the NIHR Policy Research Unit in Healthy Ageing (funding reference NIHR206119). The views expressed are those of the author(s) and not necessarily those of the NIHR or the Department of Health and Social Care.”

Authors’ response

We have included the Role of Funder statement in the Cover Letter:

Comments from Editor

3. Please include captions for your Supporting Information files at the end of your manuscript, and update any in-text citations to match accordingly. Please see our Supporting Information guidelines for more information: http://journals.plos.org/plosone/s/supporting-information [journals.plos.org].

Authors’ response

We have included the captions of the Supporting Information at the end of our manuscript and ensure the in-text citation to match accordingly. 

Comments from Editor

Authors’ response

We have reviewed our references list to ensure that it is complete and correct. We have included the additional reference in the Cover Letter and the Response to Reviewer document.

Comments from Reviewer #1

 This study used information from 7,656 adults aged 60 and older participating in the English Longitudinal Study of Ageing (ELSA, waves 6-8). Care status was assessed through received care and self-reported unmet care needs, while frailty was measured using a frailty index. Competing-risk regression analysis was used (with death as a potential competing risk), adjusted for demographic and socioeconomic confounders. Around a quarter of the participants received care, of which approximately 60% received low levels of care, while the rest had high levels of care. Older people who received low and high levels of care had a higher risk of unplanned admission independent of frailty status. Unmet need for care was not significantly associated with an increased risk of unplanned admission compared to those receiving no care. Older people in receipt of care had an increased risk of hospitalization due to falls but not fractures, compared to those who received no care after adjustment for covariates, including frailty status. Conclusions: Care receipt increases risk of hospitalization substantially, suggesting this is a group worthy of prevention intervention focus. The manuscript is interesting. However, I have a couple of questions about the frailty measurements. In Frailty index used in the present study, sarcopenia and nutritional status do not seem to be considered. It should be a limitation of the study. In frailty evaluation, both parameters are frequently included in the frailty assessment tool. Please see and discuss Abete P et al. The Italian version of the "frailty index" based on deficits in health: a validation study. Aging Clin Exp Res. 2017 Oct;29(5):913-926.

Authors’ response

Thank you for the input. We have included the exclusion of sarcopenia and nutritional status in constructing the frailty index in this manuscript in the Limitation section:

Finally, the frailty index constructed in this study did not include the diagnosis of sarcopenia and nutritional status due to the unavailability of the information in ELSA. The Italian frailty index, for instance, includes the nutritional index and provides good reliability and validity in predicting mortality, disability and hospitalisation [48]. Future research may include sarcopenia, nutritional status, and other geriatric assessments in constructing a frailty index to allow for a more comprehensive assessment of an older adult’s health.

We further added a reference in our reference list:

48. Abete P, Basile C, Bulli G, Curcio F, Liguori I, Della-Morte D, et al. The Italian version of the “frailty index” based on deficits in health: a validation study. Aging Clinical And Experimental Research. 2017;29:913-926. doi: 10.1007/s40520-017-0793-9.

Comments from Reviewer #2

Reviewer #2: This study aimed to evaluate how care receipt and unmet need for care among older people with different frailty status are associated with the risk of unplanned admission to the hospital for any cause and for conditions associated with frailty, specifically falls and fractures. This study used information from 7,656 adults aged 60 and older participating in the English Longitudinal Study of Ageing (ELSA) waves 6-8. Care status was assessed through received care and self-reported unmet care needs, while frailty was measured using a frailty index. Competing-risk regression analysis was used (with death as a potential competing risk), adjusted for demographic and socioeconomic confounders. Around a quarter of the participants received care, of which approximately 60% received low levels of care, while the rest had high levels of care. Older people who received low and high levels of care had a higher risk of unplanned admission independent of frailty status. Unmet need for care was not significantly associated with an increased risk of unplanned admission compared to those receiving no care. Older people in receipt of care had an increased risk of hospitalization due to falls but not fractures, compared to those who received no care after adjustment for covariates, including frailty status.

The study is based on a large sample size and information derived from the study are relevant for English health system. The continuity of care should be ensured by community care intervention, and I’m absolutely agree that intervention should be based on several factors such as comorbidity, frailty status and social support. [Mazzella, F., Cacciatore, F., Galizia, G., Della-Morte, D., Rossetti, M., Abbruzzese, R., et al. (2010). Social support and long-term mortality in the elderly: role of comorbidity. ARCHIVES OF GERONTOLOGY AND GERIATRICS, 51(3), 323-328] The unmet need (social and medical) is probably one of the main determinants on quality of life and appropriate health services use.

I found the study of interest. Tables should be improved and simplified. I suggest that it might be beneficial to consider adding more information to the figure legend.

Authors’ response

Thank you for the input. We have added the discussion and reference to support our statement that intervention should be based on several factors, such as comorbidity, frailty status and social support:

Prior study shows that low social support is associated with long-term mortality among older people [50].

50. Mazzella F, Cacciatore F, Galizia G, Della-Morte D, Rossetti M, Abbruzzese R, et al. Social support and long-term mortality in the elderly: role of comorbidity. Archives of Gerontology and Geriatrics. 2010;51(3):323-328. doi: 10.1016/j.archger.2010.01.011.

We have further improved and simplified Tables 1 and 2.

---

## [Decision Letter · Decision Letter 1]

26 Jun 2024

The association between frailty, care receipt and unmet need for care with the risk of hospital admissions

PONE-D-24-05107R1

Dear Dr. MAHARANI,

We’re pleased to inform you that your manuscript has been judged scientifically suitable for publication and will be formally accepted for publication once it meets all outstanding technical requirements.

Kind regards,

Pasquale Abete

Academic Editor

PLOS ONE

Additional Editor Comments (optional):

No further comments

Reviewers' comments:

Reviewer's Responses to Questions

**Comments to the Author**

1. If the authors have adequately addressed your comments raised in a previous round of review and you feel that this manuscript is now acceptable for publication, you may indicate that here to bypass the “Comments to the Author” section, enter your conflict of interest statement in the “Confidential to Editor” section, and submit your "Accept" recommendation.

Reviewer #1: All comments have been addressed

Reviewer #2: All comments have been addressed

2. Is the manuscript technically sound, and do the data support the conclusions?

Reviewer #1: Yes

Reviewer #2: Yes

3. Has the statistical analysis been performed appropriately and rigorously? 

Reviewer #1: Yes

Reviewer #2: Yes

4. Have the authors made all data underlying the findings in their manuscript fully available?

Reviewer #1: Yes

Reviewer #2: Yes

5. Is the manuscript presented in an intelligible fashion and written in standard English?

Reviewer #1: Yes

Reviewer #2: Yes

6. Review Comments to the Author

Reviewer #1: Manuscript has been improved. The revisions have enhanced the overall clarity of the work and strengthened the discussion of the topic.

Reviewer #2: The manuscript is improved and all queries were discussed.I found the manuscript suitable for publication

7. PLOS authors have the option to publish the peer review history of their article (what does this mean?). If published, this will include your full peer review and any attached files.

Reviewer #1: No

Reviewer #2: No

---

## [Editor Report · Acceptance letter]

8 Aug 2024

PONE-D-24-05107R1 

PLOS ONE

Dear Dr. Maharani, 

I'm pleased to inform you that your manuscript has been deemed suitable for publication in PLOS ONE. Congratulations! Your manuscript is now being handed over to our production team.

Kind regards, 

on behalf of

Prof. Pasquale Abete 

Academic Editor

PLOS ONE